# The Effect of Different Factors on Poly(lactic-co-glycolic acid) Nanoparticle Properties and Drug Release Behaviors When Co-Loaded with Hydrophilic and Hydrophobic Drugs

**DOI:** 10.3390/polym16070865

**Published:** 2024-03-22

**Authors:** Lianguo Wang, Pei Wang, Yifan Liu, Muhammad Atae Mustafa Mahayyudin, Rong Li, Weilun Zhang, Yilan Zhan, Zhihua Li

**Affiliations:** 1School of Stomatology, Jiangxi Medical College, Nanchang University, Nanchang 330006, China; guoguo19990713@163.com (L.W.); lyf070626@163.com (Y.L.); 413009220060@email.ncu.edu.cn (M.A.M.M.); lirong012@email.ncu.edu.cn (R.L.); 4207122044@email.ncu.edu.cn (W.Z.); 4207120011@email.ncu.edu.cn (Y.Z.); lwlq323@163.com (Z.L.); 2Jiangxi Province Key Laboratory of Oral Biomedicine, Nanchang 330006, China; 3Jiangxi Province Clinical Research Center for Oral Diseases, Nanchang 330006, China

**Keywords:** PLGA NPs, organic phase, hydrophilic drug, hydrophobic drug, BSA

## Abstract

Poly(lactic-co-glycolic acid) nanoparticles (PLGA NPs) are versatile drug nanocarriers with a wide spectrum of applications owing to their extensive advantages, including biodegradability, non-toxic side effects, and low immunogenicity. Among the numerous nanoparticle preparation methods available for PLGA NPs (the hydrophobic polymer), one of the most extensively utilized preparations is the sonicated-emulsified solvent evaporation method, owing to its simplicity, speed, convenience, and cost-effectiveness. Nevertheless, several factors can influence the outcomes, such as the types of concentration of the surfactants and organic solvents, as well as the volume of the aqueous phase. The objective of this article is to explore the influence of these factors on the properties of PLGA NPs and their drug release behavior following encapsulation. Herein, PLGA NPs were fabricated using bovine serum albumin (BSA) as a surfactant to investigate the impact of influencing factors, including different water-soluble organic solvents such as propylene carbonate (PC), ethyl acetate (PA), and dichloromethane (DCM). Notably, the size of PLGA NPs was smaller in the EA group compared to that in the DCM group. Moreover, PLGA NPs showed excellent stability, ascribed to the presence of the BSA surfactant. Furthermore, PLGA NPs were co-loaded with varying concentrations of hydrophilic drugs (doxorubicin hydrochloride) and hydrophobic drugs (celecoxib), and exhibited pH-sensitive drug release behavior in PBS with pH 7.4 and pH 5.5.

## 1. Introduction

Owing to its commendable biodegradability, biocompatibility, and minimal toxicity, poly lactic-co-glycolic acid (PLGA) has received approval from the US Food and Drug Administration (FDA) for its utilization as a drug carrier, as well as for surgical sutures and cardiovascular stents [1,2]. PLGA nanoparticles (PLGA NPs) as drug carriers offer notable benefits in the pharmaceutical industry [3,4]. Numerous techniques have been employed for the synthesis of PLGA NPs, wherein the emulsification method is the most widely used preparation method [5,6]. The particle size of PLGA NPs produced through emulsification–solvent evaporation is influenced by various factors, such as the emulsifier type and the volume ratio between the organic and aqueous phases [7,8]. Indeed, its primary determinants are the characteristics of the emulsion droplets [9,10,11,12]. Surfactants, such as bovine serum albumin (BSA) [13] and polyvinyl alcohol (PVA) [14,15], are incorporated into the continuous phase to inhibit the polymerization of emulsion droplets. This is equivalent to partially eliminating the interface between the two phases, which lowers the surface tension and surface free energy. The size of nanoparticles may diminish as the concentration of surfactant or stabilizer increases [16,17]. Numerous studies have examined the impact of individual factors on the synthesis of PLGA NPs, yet there is a paucity of research on the collective influence of multiple factors on PLGA NPs. To elucidate the multifactorial effects on PLGA NPs, researchers must conduct comprehensive investigations. Furthermore, while PLGA is commonly utilized for encapsulating hydrophobic drugs, the significance of loading two drugs with contrasting properties (hydrophilia and hydrophobicity) should not be overlooked.

Typically, the emulsification–solvent evaporation method involves dispersing and dissolving drugs in a polymer organic solution or continuous phase, with the addition of a surfactant as a stabilizer, resulting in the formation of an emulsion. Next, this drug-added polymer emulsion is combined with an aqueous solution of a surfactant to form a water-in-oil (W/O) emulsion. Nanoparticles are subsequently created using an ultrasound-breaking apparatus. Finally, a drug-containing polymer solution is introduced into the continuously stirred phase to create a stable emulsion. The solvent in the dispersed phase is initially diffused into the continuous phase, followed by complete evaporation at the water/gas interface and, ultimately, solidification into nanoparticles [18,19].

Doxorubicin hydrochloride, an anti-tumor antibiotic, exerts its inhibitory effects on RNA and DNA synthesis. It demonstrates a potent inhibitory effect on RNA, possesses a broad spectrum of anti-tumor activity, and is associated with significant side effects. This compound is characterized by its orange-red loose lump or powder form and high solubility in water. Celecoxib, a selective cyclooxygenase-2 (COX-2) inhibitor, is utilized as an anti-inflammatory and analgesic agent in the management of osteoarthritis and rheumatoid arthritis, which is not easily soluble in water. PLGA NPs facilitate the gradual release of drugs, thereby extending the duration of drug action and eliminating the need for frequent administration [20,21,22,23,24]. The utilization of PLGA NPs enhances the stability and bioavailability of drugs [25,26]. PLGA undergoes complete degradation within the living organism, resulting in the production of lactic acid and glycolic acid, which are subsequently eliminated through the tricarboxylic acid cycle in the form of water and carbon dioxide [27,28]. Moreover, the metabolic rate of PLGA can be regulated to achieve a satisfactory half-life period. Additionally, PLGA containing a higher proportion of lactic acid exhibits reduced hydrophilicity, lower water absorption, and, consequently, slower degradation rates [29]. Consequently, PLGA has garnered extensive attention as a carrier of pharmaceutical substances in recent years.

Herein, the effects of three types of organic solvents, various concentrations of surfactant, and volumes of aqueous phase on PLGA NPs were investigated using the sonicated-emulsified solvent evaporation method. Initially, the organic phase containing PLGA was dropwise added into the aqueous phase containing the surfactant (bovine serum albumin, BSA). BSA is an ionic surfactant and is very stable in the physiological environment, so it is suitable for use in the biomedical field. In addition, there are many chemical groups on BSA, and the use of modification is also more conducive to the loading of hydrophilic drugs. Then, the emulsion acquired via sonication was diluted and mixed to obtain a volatilizing organic solvent (Figure 1a). The size, zeta potential, and other physiochemical characteristics of PLGA NPs were determined to confirm the relevant parameters for the sonicated-emulsified solvent evaporation method. Afterward, to co-load hydrophilic and hydrophobic drugs, celecoxib (Ceb) was dissolved in the organic phase, whilst doxorubicin hydrochloride (Dox) was dissolved in the aqueous phase during the preparation of PLGA NPs. Ceb was physically mixed with PLGA and loaded into the nanoparticles, while Dox was loaded into the nanoparticles via electrostatic interactions and π-π conjugation with BSA (Figure 1b). Moreover, the oil–water partition coefficients of Dox and Ceb were 1.50 and 4.21, respectively. Following this, drug release behaviors were monitored under different acidic conditions to simulate the physiological and lysosomal environments. Moreover, the physiochemical characteristics of drug-loading PLGA nanoparticles were also examined.

## 2. Materials and Methods

### 2.1. Materials

The materials used were poly lactic-co-glycolic acid (PLGA, lactide:glycolide 75:25, Mw 4000–15,000, Sigma Aldrich, Shanghai, China, ≥99.9% purity), Bovine serum albumin, (BSA, Aladdin, Shanghai, China, 96% purity), ethyl acetate (EA, Sinopharm Chemical Reagent Co., Ltd., Shanghai, China, analytically pure), propylene carbonate (PC, Aladdin, 99.7% purity), dichloromethane (DCM, Sinopharm Chemical Reagent Co., Ltd., analytically pure), doxorubicin hydrochloride (Dox, milunbio, Dalian, China, ≥98% purity), Celebrex (Ceb, milunbio, ≥98% purity), sodium hydroxide (Sinopharm Chemical Reagent Co., Ltd., analytically pure), Sodium dodecyl sulfate (SDS, Sigma Aldrich, ≥99.9% purity), methanol (Sinopharm Chemical Reagent Co., Ltd., analytically pure), and Methyl sulfoxide (Sinopharm Chemical Reagent Co., Ltd., analytically pure).

### 2.2. The Synthesis of PLGA NPs and Loading with Dox and Ceb

Firstly, PLGA NPs were synthesized by applying the emulsified solvent evaporation method, as described in previous studies [30] (Fan et al., 2017). Briefly, 20 mg of PLGA was dissolved in 1 mL of different organic solvents (EA, PC, or DCM) to compare the effects of the solvents on nanoparticles. BSA aqueous solution (concentration listed in Table 1) was then introduced, and the mixture was sonicated for 60 s. Thereafter, the emulsion was added dropwise to 30 mL ultrapure water and stirred at 800 rpm for 6 h. Finally, PLGA NPs were collected using centrifugation at 10,000 rpm for 10 min and washed for 3 times, lyophilized, and stored at 4 °C for the ensuing analyses.

The oil phase of PLGA was supplemented with 1 mg Dox for preparing PLGA NPs loaded with Dox (PD NPs), and sonicated for 30 s with the power of 1000 W (Scientz IID, Scientz, Suzhou, China). Similarly, PLGA NPs loaded with Ceb (PC NPs) were prepared by dissolving the polymer in an organic solvent with 1 mg Ceb. To prepare PLGA NPs co-loaded with Dox and Ceb (PDC NPs), the oil phase of PC NPs was added to 50 μL of 20 μg/mL Dox solution. The remaining procedures were identical.

### 2.3. Physiochemical Characterizations

The structure and morphology of PLGA NPs, PD NPs, PC NPs, and PDC NPs were characterized under a transmission electron microscope (TEM, H-7650, Hitachi, Tokyo, Japan). The particle size and zeta potential of the three kinds of nanoparticles dispersed into water and PBS were measured using a Zetasizer (NanoBrook Omni, Brookhaven, GA, USA). The samples were qualitatively analyzed using a Fourier-Transform Infrared Spectrometer (FTIR, Nicolet iS 50, Thermo Fisher Scientific, Waltham, MA, USA). The crystal structures of PD NPs, PC NPs, and PDC NPs were evaluated using X-ray diffraction (XRD, D8 Advance, Bruker AXS, Karlsruhe, Germany). The residue of organic solvent was determined using a gas chromatograph (GC-2014C, Shimadzu, Kyoto, Japan).

### 2.4. The BSA Assay by BCA Kit

To examine the residual amount of BSA in PLGA NPs prepared with different organic phases and concentrations, 3 mg of PLGA NPs was dissolved in 1 mL of DMSO using sonication. Then, the solution was diluted in 10 mL water using a colorimetric tube. The residual amount of BSA was thereupon measured based on the calibration curve of BSA using a BCA protein assay kit from KeyGen Biotechnololy Co., Ltd. (Nanjing, China). In short, 200 µL BCA reagent and 20 µL sample solution were incubated in a 96-well plate at 45 °C for 1 h. Absorbance was detected using a microplate reader (Infinite 200 Pro, Tecan, Grödig, Austria) at 562 nm.

### 2.5. The Measure of the Drug Loading and Encapsulation Efficiency and In Vitro Drug Release

Initially, the drug loading content and encapsulation efficiency were determined using ultraviolet absorption spectrophotometry. Briefly, after the preparation of PD NPs, PC NPs, and PDC NPs, the residual content of the drug molecule in the supernatant was measured. The absorption values of the sample solution were measured at a wavelength of 480 nm for Dox, using an ultraviolet–visible (UV–Vis) spectrophotometer (V-670, Jasco, Tokyo, Japan), and at 254 nm for Ceb, using high-performance liquid chromatography (HPLC, 1260 Infinity II, Agilent Japan, Tokyo, Japan). The column was Hypersil ODS C18 (250 mm × 4.6 mm, 5 µm) and the mobile phase was methyl alcohol–water (85:15). The drug loading and encapsulation efficiency were calculated based on the standard curve:(1)$$Loading\;content=\frac{m1-m2}{m}\times 100\%$$
(2)$$Encapsulation\;efficiency=\frac{m1-m2}{m1}\times 100\%$$
where *m*1 represents the mass of the total drug, *m*2 denotes the mass of the drug in the centrifugal supernatant, and *m* stands for the total mass of the sample.

Then, the 5 mg samples dispersed into 5 mL PBS with 1% tween 80 were placed into dialysis bags (interception molecular weight: 8000 Da), which were then immersed in 25 mL PBS (pH = 5.5, pH = 7.4) at 37 °C with continuous shaking at 100 rpm in an orbital shaker. At predetermined intervals (0.5, 1, 2, 4, 8, 12, 24… h), 3 mL buffer solution was discarded and replaced with fresh buffer solution. Lastly, the ultraviolet absorption values of the obtained samples were detected using HPLC and a UV–vis spectrophotometer at wavelengths of 480 nm and 254 nm. The standard plots of Dox and Ceb were constructed under identical conditions. The in vitro drug release curves of PD NPs, PC NPs, and PDC NPs were plotted using standard curve calculation.

### 2.6. Statistical Analysis

All data were expressed as the mean ± standard deviation (n ≥ 3). The statistical significance of the data was calculated via a one-way analysis of variance (ANOVA) followed by Tukey’s test. A *p*-value of <0.05 was considered statistically significant (* *p* < 0.05, ** *p* < 0.01, *** *p* < 0.001).

## 3. Results and Discussion

### 3.1. Size of PLGA NPs

To observe the effects of different types of organic phase solvents, surfactant concentrations, and rates between the organic and aqueous phases, the size and zeta potential of PLGA NPs were detected using Zetasizer through dynamic light scattering following their preparation via the ultrasonic emulsification and solvent evaporation method, as illustrated in Figure 2 and Appendix A. Evidently, the size of PLGA NPs was obviously different in the three types of organic solvents (DCM, EA, and PC) (Figure 2a). The results showed that there was little effect on the size of PLGA NPs when the BSA concentration ranged from 0.5% to 3% in DCM. Nevertheless, there is a large deviation in the preparation of PLGA NPs using DCM, indicating that the stability of the DCM preparation of PLGA NPs is not as good as the other two organic solvents. In addition, the size of PLGA NPs remained relatively consistent using different BSA concentrations, indicating that the concentration of the surfactant might be saturated and consequently had no impact on PLGA NP synthesis [31]. Moreover, the sizes of PLGA NPs when using DCM as the organic solvent were about 160 nm at various concentrations of BSA, revealing that the concentration of BSA has minimal effect on particle size in the DCM group. This may be attributed to the low water solubility of the organic phase, where the concentration of the surfactant (BSA) has limited influence on particle size, or when concentrations vary significantly to yield noticeable differences. Contrastingly, the sizes of PLGA NPs using EA and PC as organic solvents were roughly 100 nm and 160 nm at a BSA concentration of 0.5% and about 80 nm and 100 nm at the other BSA concentrations, respectively. It is worthwhile pointing out that the size of nanoparticles obtained from the organic phase of propylene carbonate was not the smallest, which contradicts the findings of other reports. It was reported that nanoparticles prepared from PC with partial water solubility were smaller than those prepared from EA due to the slightly higher water solubility of PC [17]. However, in our study, nanoparticles prepared using EA as the organic solvent exhibited a smaller size. This can be ascribed to the relatively high viscosity of PC, which consequently increases the system viscosity during the preparation process, leading to the larger particle sizes of PLGA NPs.

On the one hand, the size of PLGA NPs using DCM also fluctuated with varying volumes of aqueous phase at a BSA concentration of 2% (Figure 2b). On the other hand, the size of PLGA NPs using EA as the organic solvent decreased but consistently increased with increasing volumes of the aqueous phase when PC was used. As depicted in Figure 2, the EA organic phase yielded nanoparticles with a smaller particle size compared to the other two organic phases. Specifically, the smallest nanoparticles were obtained using 1% BSA and 5 mL water, 2% BSA and 4 mL water, and 3% BSA and 4 mL water, measuring 122.22 ± 0.47 nm, 126.32 ± 0.75 nm, and 117.22 ± 0.42 nm, respectively. Consequently, 2% BSA and 4 mL water were selected for the formulation of PLGA NPs. We hypothesized that an increase in the proportion of the water phase would result in a corresponding decrease in the particle size of PLGA NPs. This assumption was based on the notion that a greater amount of aqueous phase would prevent contact with the organic phase, thereby leading to smaller particle sizes [25,28]. Nevertheless, the results are different. One potential explanation is that DCM, being a non-water-soluble organic phase, is not significantly affected by alterations in water phase volume, resulting in minimal impact on particle size [32,33]. The primary factors influencing particle size alteration are ultrasonic intensity and BSA concentration. In contrast, EA and PC exhibit solubility in water at approximately 10% and 20%, respectively. Consequently, an increase in water phase volume led to the enhanced dissolution of EA, thereby resulting in a smaller PLGA NP particle size following ultrasound treatment. However, the solubility of PC surpassed that of EA. The reason behind the increase in particle size with the expansion of the water phase volume primarily lies in the elevated viscosity of PC when dissolved in water, leading to a rise in the overall system viscosity, consequently resulting in larger particle sizes.

### 3.2. Zeta Potential of PLGA NPs

To further evaluate surface characterization, PLGA NPs were not dispersed only in water but also in PBS to mimic the physiological (high-electrolyte) environment. As displayed in Figure 3a,b, the zeta potentials of PLGA NPs were investigated in an aqueous solution. It is evident that the absolute value of the zeta potential of PLGA NPs prepared in DCM, which was below 30 mV, was lower than in EA and PC. This discrepancy suggests that the surface potential of PLGA nanoparticles prepared using DCM was significantly smaller and influenced interparticle interactions. The significance of zeta potential is that its value is related to the stability of colloidal dispersion. The smaller the dispersed particle and the higher the absolute value (positive or negative) of the Zeta potential, the more stable the system, that is, the dissolution or dispersion can resist aggregation [34].

In addition, to further observe the stability of PLGA NPs in high-salt solutions, PLGA NPs were dissolved in PBS to measure zeta potentials (Figure 3c,d). The results revealed that the magnitude of the zeta potential of PLGA NPs in PBS was significantly lower compared to that in the aqueous solution. Conversely, the zeta potential of PLGA NPs synthesized using EA was considerably higher than that of the other two organic solvents (DCM and PC), showcasing a disparity from the outcomes observed in water. However, this discrepancy also suggested that PLGA NPs prepared using EA exhibited favorable stability. Nanoparticles possessing a smaller overall charge and a low absolute zeta potential tend to attract other nanoparticles, thereby inducing instability within the entire system.

### 3.3. The Residual Amount of BSA in PLGA NPs

Residual BSA is a key obstacle to the stability of PLGA NPs and the loading content of hydrophilic drugs. To detect the BSA rate, PLGA NPs were dissolved in DMSO and then assayed using a BCA kit. The calibration curve of BSA was calculated as y = 0.0023x − 0.0032, R^2^ = 0.9994 (Appendix A), demonstrating its excellent linear relationship with the regression equation. As delineated in Figure 4a,b, the EA group had the highest residual amount of BSA, exceeding 10%, followed by the DCM group, while the PC group had the lowest residual amount. Notably, the residual amount of BSA varied among PLGA NPs prepared using different organic phases. Specifically, the EA group exhibited the highest residual amount, whereas the PC group displayed the lowest residual amount. This discrepancy could be attributed to the smaller size and larger surface area of nanoparticles, which facilitate the adsorption of BSA. One potential explanation for this phenomenon is the high polarity and water solubility of PC, resulting in interactions between PC and BSA during the nanoparticle preparation process. In this scenario, BSA dissolves in PC, and its ionic surfactant properties contribute to a lower BSA concentration within PC, consistent with the aforementioned outcomes.

As the concentration of BSA increased, the residue of BSA in group EA increased, whereas that in the remaining two groups decreased (Figure 4a). Interestingly, the particle size of PLGA NPs in the DCM and PC groups remained relatively stable. This could potentially be ascribed to the excess concentration of BSA, which consequently promoted electrostatic repulsion between BSA and PLGA NPs, thereby resulting in a notable reduction of BSA absorbed on the coating of PLGA NPs. The amount of BSA residue in group EA increased with an increase in water phase volume, whereas that of the other two groups decreased (Figure 4b). Additionally, the particle size in the EA group decreased, so the specific surface area increased to increase the residual amount of BSA, whereas that in the DCM group remained relatively unchanged, and that in the PC group increased.

### 3.4. TEM Images of PLGA NPs

The analysis presented in Figure 5 demonstrates that the nanoparticles exhibit a solid spherical structure with a continuous distribution and no signs of aggregation, signaling that the prepared PLGA nanoparticles possess outstanding dispersion characteristics. Furthermore, the particle size, approximately 100 nm, was in agreement with the results obtained from both TEM observation and DLS measurement, ensuring consistency between the two techniques. After freeze-drying, the presence of DCM and EA in PLGA NPs was undetectable owing to their low boiling points in the organic solvent residing assays. Conversely, a residual quantity of PC, approximately 0.89%, was noted owing to its elevated boiling point.

### 3.5. The Characteristics of Drug-Loaded PLGA NPs

To observe the change in PLGA NPs after loading drugs (Dox and Ceb), the characteristics of PD NPs, PC NPs, and PDC NPs were investigated. Figure 6a depicts the mean particle size of nanocarriers containing Dox and Ceb. Nanocarriers co-loaded with Dox and Ceb had an average particle size of 85.6 ± 7.8 nm. The zeta potential of PD NPs, PC NPs, and PDC NPs were measured as −30.8 ± 1.2 mV, −30.5 ± 0.3 mV, and −30.4 ± 0.5 mV, respectively. These values fall within the range of −20 to −40 mV (Figure 6b), signifying that the nanocarriers possessed a higher surface potential and electrostatic potential energy between particles. Overall, these findings suggested that the prepared nanomedical carrier system demonstrated favorable stability. As shown in Figure 6c, the FTIR spectrum of PDC NPs reveals the presence of characteristic peaks, including the carbonyl group at 1750 cm^−1^ representing PLGA, peaks at 3300 cm^−1^ and 1580 cm^−1^ corresponding to BSA, and peaks at 3300 cm^−1^, 1725 cm^−1^ and 1250 cm^−1^ associated with Dox, indicating the successful preparation of PD NPs and PC NPs. The X-ray diffraction (XRD) spectrum of nanoparticles is depicted in Figure 6d. Comparing the XRD spectra uncovered significant alterations, with numerous peaks in Dox and Ceb absent in PD NPs, PC NPs, and PDC NPs. This discrepancy suggests a modification in the crystal structure subsequent to drug loading. The baseline in PC NPs demonstrated irregularities, indicating a mixed crystal composition.

The TEM observation of PD, PC, and PDC nanoparticles showed the following morphological characteristics. From the TEM images (Figure 7), the fabricated PLGA NPs loaded with Dox and Ceb through physical embedding and electrostatic adsorption possessed a spherical structure with a size of about 100 nm. Moreover, the hydrophilic and hydrophobic drugs (Dox and Ceb) had no impact on the morphological characteristics of PLGA NPs, including their size (≈100 nm). Additionally, the presence of BSA, a surfactant with strong electrostatic repulsion properties, did not induce PDC NP agglomeration (Figure 7c). Overall, our nanoformulation resulted in excellent dispersibility, a critical index in the field of nanobiomaterials for biomedical applications.

### 3.6. In Vitro Drug Release Investigation

To measure the cumulative release rate of drugs, the drug contents and encapsulation efficiency were calculated (Figure 8a and Appendix A). The drug contents and encapsulation efficiency of Dox in PD NPs, Ceb in PC NPs, Dox and Ceb in PDC NPs were about 4% and 85%, 1.8% and 57%, about 4% and 80%, and 1% and 41%, respectively, according to the calibration curve for Dox and Ceb. The drug loading of Ceb was significantly lower than that of Dox, possibly attributable to the hydrophobicity of Ceb in the aqueous phase. Additionally, the fed dosages of Dox and Ceb were limited to load into PLGA NPs to preserve the physiochemical properties and structures of PLGA-based nanoparticles, which was the reason of the low drug contents for PD NPs, PC NPs, and PDC NPs.

To further investigate the release of Dox and Ceb at different conditions, PBS with pH 7.4 and 5.5 was chosen to stimulate the physiological condition and lysosomal condition. The results highlighted that PD NPs only loaded with Dox presented a sustainable and controlled release (Figure 8b). The cumulative release rate of Dox was approximately 20% higher in PBS with pH 7.4 than 5.5 within 72 h, which could be attributed to the neutralization of H^+^ from doxorubicin hydrochloride under a mildly alkaline environment, with the deficiency of H^+^ facilitating its release from PD NPs. However, it is worth noting that Ceb, unlike Dox (a water-soluble drug), lacks any acidic-reactive functional groups. On the contrary, the sustained release of Ceb (neutral drug) from PC NPs in PBS with pH 7.4 and pH 5.5 was parallel (Figure 8c). The cumulative release of Ceb (hydrophobic drug) was approximately 50% for 72 h through the diffusion process, detected using HPLC with a peak time of 4.5 min at an absorption peak of 254 nm. Further, the cumulative release of hydrophilic and hydrophobic drugs (Dox and Ceb) within PDC NPs under different conditions (pH 7.4 and 5.5) was examined (Figure 8d). Intriguingly, the release of Ceb presented pH-sensitive prosperity in PDC NPs, different from PC NPs. Conversely, when Ceb was independently loaded in PC NPs, no pH-responsive drug release was observed, indicating that the release of Dox accelerated the release of Ceb. In other words, the release of Dox might induce a structural collapse in PDC NPs, thereby expediting the release of Ceb. The results demonstrated that the release rate of Dox was the highest at pH 7.4, whereas Ceb displayed the slowest release rate at pH 5.5.

## 4. Conclusions

In summary, the optimal conditions for manufacturing PLGA NPs via the sonicated-emulsified solvent evaporation method were successfully assayed herein, including the type of organic solvent, surfactant concentration, and the volume of the aqueous phase. Our observations uncovered that the size of PLGA NPs was significantly impacted by the type of organic solvent, with remarkably smaller sizes in water-soluble organic solvents (EA and PC) compared with a water-insoluble organic solvent (DCM). The size of PLGA NPs was the smallest in the EA group, which may be due to its favorable solubility, low viscosity, and boiling point. Nevertheless, variations in BSA concentration and the volume of the aqueous phase had minimal impact on PLGA NP sizes. The ultimate choice for our designed PLGA NPs was 2% BSA concentration, 4 mL aqueous solution, and EA as an organic solvent. All parameters conjointly demonstrated excellent stability in water and PBS, maintained a nanosphere structure, and retained the highest residual amount of BSA, thereby ensuring stability and drug content. Furthermore, a hydrophobic drug (Ceb) and a hydrophilic drug (Dox) were co-loaded to address clinical demands for multi-drug applications. Notably, the drug-loading content of Dox were higher than that of Ceb, suggesting a significant interaction between BSA and Dox. Moreover, PDC NPs exhibited stimuli-responsive drug release in different acidic solutions, making them suitable for various clinical scenarios.

## Figures and Tables

**Figure 1 polymers-16-00865-f001:**
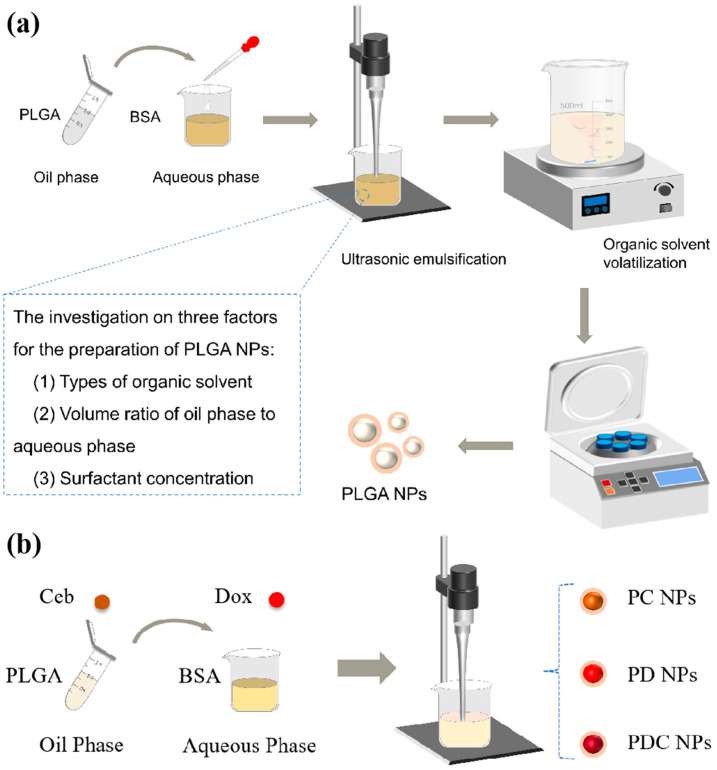
(**a**) Schematic illustration of (**a**) the step-by-step preparation of PLGA NPs at different synthetic factors and (**b**) the drug loading process including PLGA NPs loaded with Dox (PD NPs), PLGA NPs loaded with Ceb (PC NPs), and PLGA NPs loaded with Dox and Ceb (PDC NPs).

**Figure 2 polymers-16-00865-f002:**
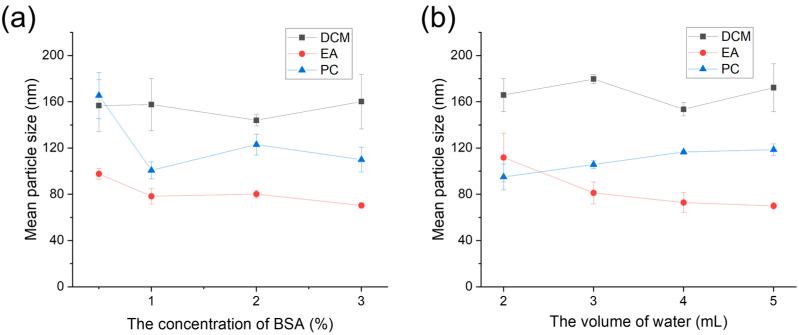
The size of PLGA NPs prepared in different organic phases (DCM, EA, and PC) through the emulsified solvent evaporation method under (**a**) various BSA concentrations and (**b**) various volumes of water phase.

**Figure 3 polymers-16-00865-f003:**
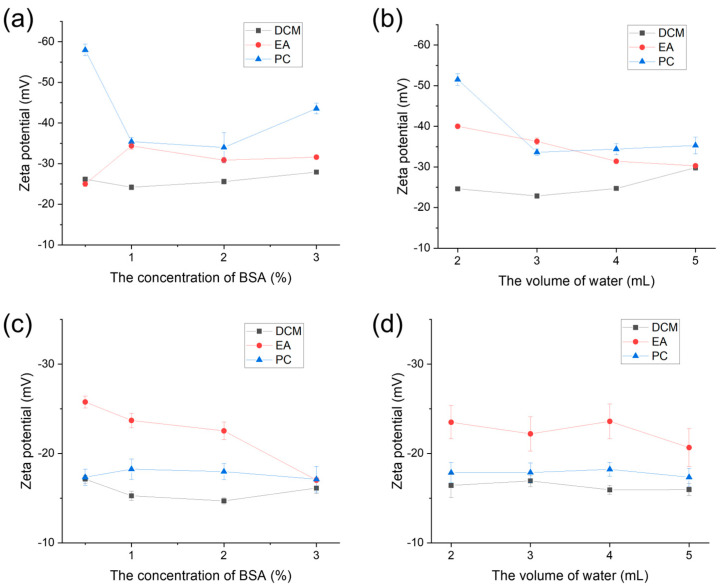
Zeta potential of PLGA NPs prepared in different organic phases (DCM, EA, and PC) through the emulsified solvent evaporation method (**a**) under various BSA concentrations and (**b**) various volumes of aqueous phase dispersed in water and PBS (**c**,**d**).

**Figure 4 polymers-16-00865-f004:**
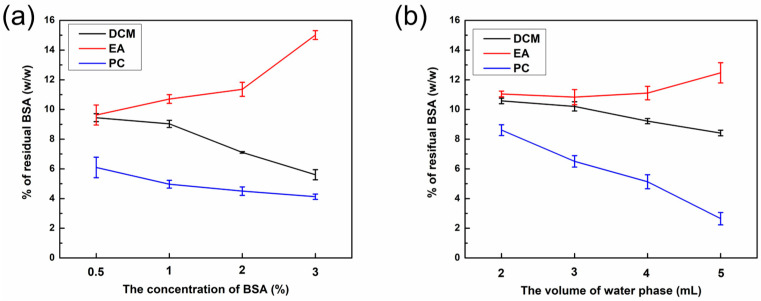
(**a**) The residual rate of BSA in PLGA NPs with the various volumes of water phase and different organic phases (DCM, EA, and PC) with a 1% fixed concentration of BSA. (**b**) The residual rate of BSA in PLGA NPs with the various concentrations of BSA and different organic phases (DCM, EA, and PC) with 4 mL fixed volume of water phase.

**Figure 5 polymers-16-00865-f005:**
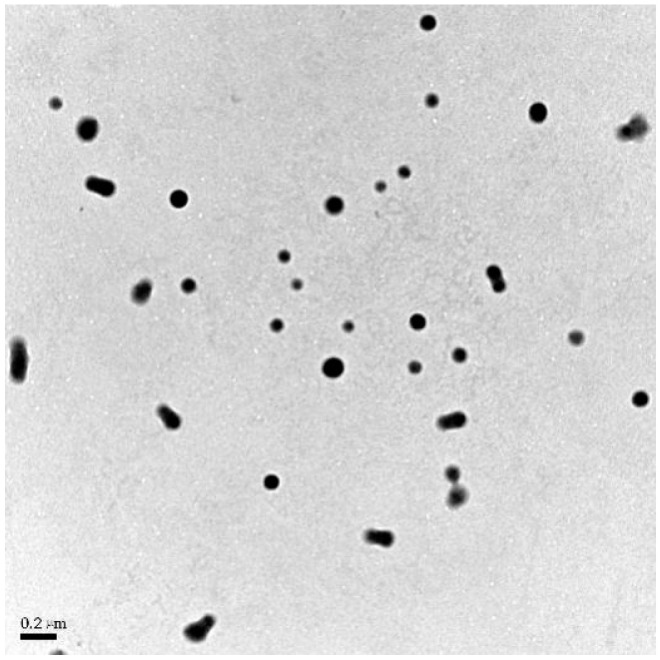
TEM images of PLGA NPs (100 kV voltage, 30 k magnification).

**Figure 6 polymers-16-00865-f006:**
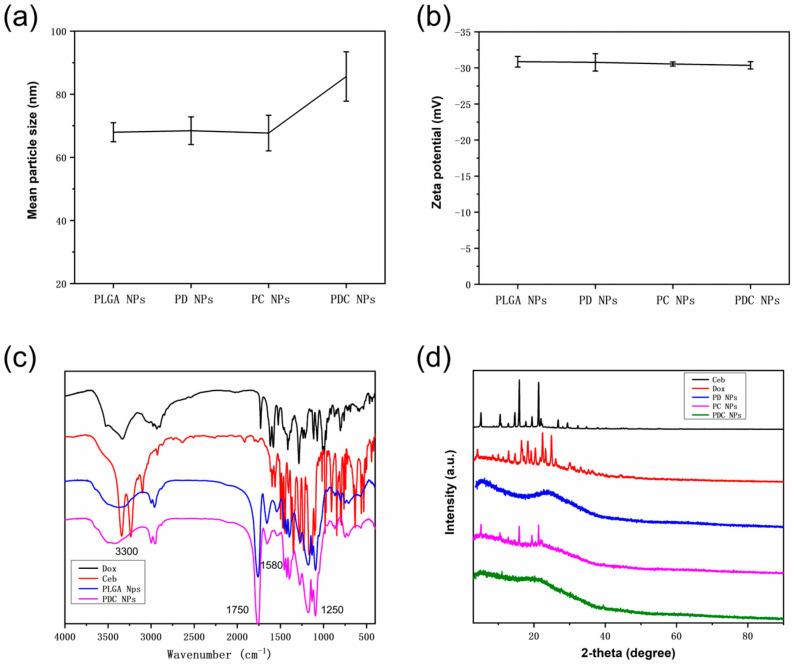
(**a**) Size, (**b**) zeta potential, (**c**) FTIR spectrum, and (**d**) XRD pattern of PLGA NPs, PD NPs, PC NPs, and PDC NPs.

**Figure 7 polymers-16-00865-f007:**
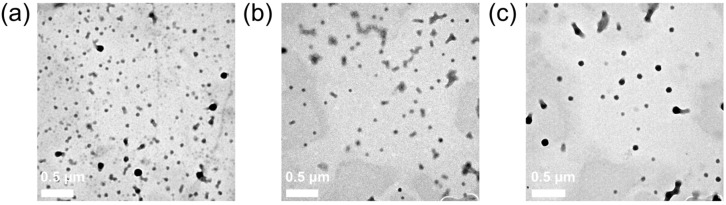
TEM images of (**a**) PD NPs, (**b**) PC NPs, and (**c**) PDC NPs.

**Figure 8 polymers-16-00865-f008:**
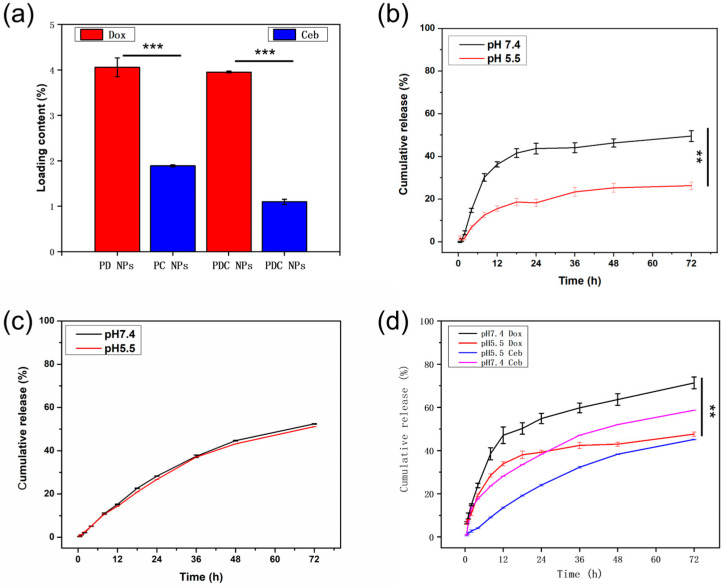
(**a**) The loading content of PD NPs, PC NPs, and PDC NPs. (**b**) The cumulative release profiles of Dox from PD NPs in PBS (pH 7.4 and 5.5). (**c**) The cumulative release profiles of Ceb from PC NPs in PBS (pH 7.4 and 5.5). (**d**) The cumulative release profiles of Dox and Ceb from PDC NPs in PBS (pH 7.4 and 5.5) ** *p* < 0.01, *** *p* < 0.001.

**Table 1 polymers-16-00865-t001:** Concentration of BSA aqueous solution.

**Concentration of BSA**	0.5%	1%	2%	3%
**BSA/mg**	20	40	80	120
**Water/mL**	4	4	4	4

## Data Availability

Data are contained within the article.

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
