# Peer review of "The Effect of Different Factors on Poly(lactic-co-glycolic acid) Nanoparticle Properties and Drug Release Behaviors When Co-Loaded with Hydrophilic and Hydrophobic Drugs"

_polymers, 2024, doi:10.3390/polym16070865_

Round 1
Reviewer 1 Report
Comments and Suggestions for Authors
The manuscript is dedicated to the current, but at the same time well-studied area of research of formation of PLGA nanoparticles containing doxorubicin by emulsion solvent evaporation method. The fast Google search with “emulsion solvent evaporation plga nanoparticles doxorubicin” key words gives about 66800 results, the same with “BSA” - 25 800 res. That is why despite the relevance of the task - the creation of long-acting anticancer drugs, the novelty of the research is not obvious. It must be thoroughly described and discussed in the introduction.
Besides, there are some additional remarks:
1. Line 38: “in the production of lactic acid and acetic acid” – not acetic, but glycolic.
2. What is the nature of the big differences in errors in Fig.2? The data look very strange. Most of the curves based only on 4 experimental points have extrema, some of them more than one. The data was not accurately compared with those obtained from TEM images analysis, for what the only mean size (100 nm) is given. The size distribution, very important parameter is not discussed. Black curve in Fig. 2(a) does not demonstrate any dependence on BSA content which is discussed in the manuscript.
3. Line 277-278. The phrase “Nanocarriers co-loaded with Dox 287 and Ceb had an average particle size of 231.74 ± 1.10 nm” contradicts with Fig.6 (a) (70-90nm).
After major revision the manuscript may be published.
Reviewer 2 Report
Comments and Suggestions for Authors
Comments:
1. The title needs to include the proper word, e.g., "...PLGA Nanoparticles characteristics/properties ..."
2. All abbreviations such as EA, DCM, BCA should be included at their first mention.
3. The aims of the study are not apparent; consider elaborating on them in the abstract and introduction sections.
4. Clarify and add the relevant reference at line 40.
5. Add reference(s) to line 86.
6. Include the full names of PC, PD, and PDC NPs in the caption of Figure 1.
7. Why were different drug concentrations (20 μg/mL Dox vs 1 mg Ceb) used?
8. Considering Celecoxib's hydrophobic nature and low solubility in PBS (log P = 3.5), how was it released in the PBS medium?
9. Add the relevant reference to lines 186-188.
10. Improve the quality and resolution of Figures 5 and 7.
11. How was the value of 0.89% calculated and obtained (lines 280-281)?
12. There is a discrepancy between the mentioned results in section 3.5 and Figures 6a and b; clarify this inconsistency.
13. In the FTIR spectrum of PDC NPs, the characteristic peaks of Ceb in the range of 3000-3500 cm-1 do not appear. What is the reason for this absence?
14. The TEM results do not align with the statement "regular spherical structure with a uniform size of about 100 nm." The prepared NPs appear non-uniform in size and morphology. Clarify this discrepancy.
Round 2
Reviewer 1 Report
Comments and Suggestions for Authors
The authors did a good job and substantially improved the Introduction, the actuality and the main goals of the research. I have no more complaints about this section. Nevertheless, the description and interpretation of the results regarding the dependence of particle sizes on various factors is questionable. As follows from Fig. 2 and the text of the manuscript, "the size of PLGA NPs fluctuated with alterations in BSA (as surfactant) concentration, which ranged from 0.5% to 3%, but the largest size was observed at a BSA concentration of 0.5% and an aqueous volume of 4 mL". Really, all the difference is beyond the specified experimental errors and sometimes fluctuates up and down (see black curve). Perhaps these fluctuations reflect an experimental error (different from the one indicated in Fig. 2) and the real dependences are not observed. Then the conclusion given above about optimal concentrations is invalid. If the authors believe that the fluctuations are the real experimental fact, this should be confirmed by supplementing with measurements at intermediate points, for example, at BSA concentration of 1.5 and 3.5%.
Reviewer 2 Report
Comments and Suggestions for Authors
The corrections have been made and are approved and accepted.
Author Response
Dear Reviewer,
Thank you for your thorough review and feedback on our manuscript. We appreciate your time and effort in helping us improve the quality of our work. Your feedback has been invaluable in enhancing the clarity and accuracy of our manuscript.
Once again, thank you for your support and assistance. We look forward to the publication of our manuscript and the opportunity to share our research with a wider audience.
Best regards,
Lianguo Wang